# ViVa: Video-Trained Value Functions for Guiding Online RL from Diverse Data

## Abstract

Online reinforcement learning (RL) with sparse rewards poses a challenge partly because of the lack of feedback on states leading to the goal. Furthermore, expert offline data with reward signal is rarely available to provide this feedback and bootstrap online learning. How can we guide online agents to the right solution without this on-task data? Reward shaping offers a solution by providing fine-grained signal to nudge the policy towards the optimal solution. However, reward shaping often requires domain knowledge to hand-engineer heuristics for a specific goal. To enable more general and inexpensive guidance, we propose and analyze a data-driven methodology that automatically guides RL by learning from widely available video data such as Internet recordings, off-task demonstrations, task failures, and undirected environment interaction. By learning a model of optimal goal-conditioned value from diverse passive data, we open the floor to scaling up and using a wide variety of data sources to model general goal-reaching behaviors relevant to guiding online RL. Specifically, we use intent-conditioned value functions to learn from diverse video and incorporate these goal-conditioned values into the reward. Our experiments show that video-trained value functions work well with a variety of data sources, exhibit positive transfer from human video pre-training, can generalize to unseen goals, and scale with dataset size.

## 1 Introduction

Many sequential decision-making tasks are naturally defined with a sparse reward, meaning the agent only receives positive signal when the goal has been achieved. Unfortunately, these sparse reward tasks are especially challenging in reinforcement learning (RL) (Sutton, 2018) since they provide no signal at intermediate states, effectively requiring exhaustive search. Practitioners often resort to collecting task-relevant prior data (Pomerleau, 1988) or hand-designing task-relevant dense reward functions (Mataric, 1994). However, manually collecting this high-quality data or defining a task-specific reward is time-intensive and not general.

To solve this problem in RL, we should guide the search procedure online towards the desired goal. This dictates the usage of some general prior informing the agent what states lead to others to direct it to the goal. Humans make use of extensive prior knowledge when attempting to accomplish tasks: for example, we know that finding a mug generally requires us to try looking in cabinets and that opening them requires interacting with the handle. We posit that this prior can in fact be learned with task-agnostic environment data and general manipulation videos to develop a sense of "how the world works." This data is easily collected in the environment or mined from the web, respectively.

To leverage both of these data types, we choose to learn from *video*, enabling the use of a myriad of datasets without needing embodiment-specific actions or task-specific rewards. The nature of video data availability on the web also allows for training on other environments. We hypothesize that learning models from various video sources will expand the data support, enabling generalization and successful goal-reaching guidance. Crucially, we elect to represent our prior as a *goal-conditioned state-value function* $V(s, g)$, that for any image $s$ and desired target $g$, estimates the temporal distance between the two states. Learning this type of model easily plugs into online RL by penalizing the predicted distance from the goal. Also, using a value-learning approach allows ingesting suboptimal reaching data, further relaxing our requirements for the training data, as opposed to other behavioral prior methods (Escontrela et al., 2023). Lastly, goal-conditioned value

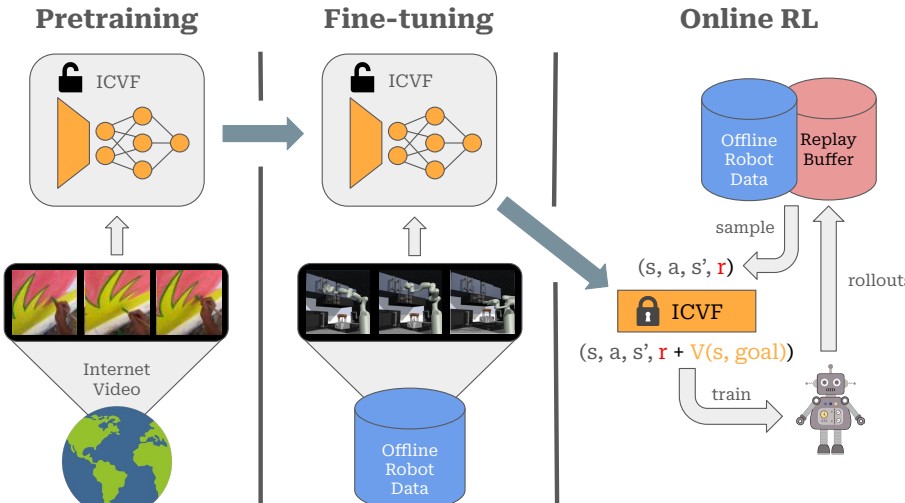

Figure 1: **Left:** ViVa uses samples from internet-scale video to learn a value-function that encodes goal-reaching priors. **Middle:** ViVa finetunes on robotics-relevant data to bring the value function into the domain of the tasks we wish to solve. **Right:** During online RL, we freeze the value function and augment the extrinsic reward with a guidance signal that captures temporal distances. We choose to include the robotics-relevant interaction data in our online pipeline to assist exploration.

functions naturally extend to multiple tasks by flexible goal specification. In essence, we desire a simply learned function that scales with widely available off-task and off-environment video data to generally inform the agent about useful states leading to the goal.

We can instantiate our method by pre-training an Intent-conditioned Value Function (ICVF) (Ghosh et al., 2023) on Internet-scale egocentric interaction data in various settings (Ego4D) (Grauman et al., 2022). We use this training to develop strong visual features as well as a priors over object manipulation and interaction outcomes. We then finetune this ICVF on environment-specific, yet task-agnostic data, to specialize the function for the setting of interest. During online RL, we provide the temporal distance estimates to the agent in the form of a reward penalty.

We observe that reformulating online sparse RL problems with **Vi**deo-trained **Va**lue functions (ViVa) shows a number of benefits. Firstly, we see generalization to new goals unseen in prior data in the Antmaze environment (Fu et al., 2021), a simple state-based control setting. Secondly, we also see improvement in performance by training on off-task data in a visual robotic simulator, RoboVerse (Singh et al., 2020). Thirdly, we see that pre-training on Ego4D can significantly improve performance but is not sufficient to solve online RL alone, necessitating some environment finetuning. Lastly, we see that ViVa improves online performance as data scale increases and can enable solving complex robotic tasks on Franka Kitchen (Gupta et al., 2019), another robotic simulator.

## 2 RELATED WORK

Solving sparse online RL problems is a difficult challenge due to the lack of reward feedback. One way to make it easier is to better explore the environment to more reliably reach the goal state and begin backing up rewards. They range from simple noisy behaviors (Haarnoja et al., 2018b) to structured behavioral priors (Ecoffet et al., 2021; Bharadhwaj et al., 2021; Kearns & Singh, 2002; Brafman & Tennenholtz, 2003). Some methods utilize intrinsic bonuses (Schmidhuber, 2010) to minimize uncertainty (Kolter & Ng, 2009; Pathak et al., 2019; Houthooft et al., 2017; Still & Precup, 2012) or to seek novelty (Burda et al., 2018; Pathak et al., 2017; Ostrovski et al., 2017; Tang et al., 2017; Bellemare et al., 2016). Unfortunately, these methods break down in complex visual environments and intricate robotic control settings due to the large state and action space.

To narrow this search, a prior is desirable to inform the agent of what states or actions to explore more. One way to do this is to inject domain knowledge into the reward function, guiding it to

the goal. This family of approaches, known as reward shaping, can accelerate learning the optimal policy (Ng et al., 1999; Mataric, 1994; Hu et al., 2020; Devlin & Kudenko, 2012; Wiewiora, 2003). However, hand-crafting these rewards does not generally scale to many tasks and is often over-designed for one domain (Jiang et al., 2020; Mahmood et al., 2018; Haarnoja et al., 2018a; Malysheva & Kudenko, 2018; Hussein et al., 2017; Brys et al., 2015). A more ideal way to have a general prior is to learn it from a wide range of available data. In this work, we explore the effect of various video data sources in providing robotics-relevant dynamics information to downstream RL.

Many methods elect to use this cheap video data to learn a rich image representation through reconstruction objectives (Xiao et al., 2022), constrastive learning (Nair et al., 2022), value-functions (Bhateja et al., 2023), or predictive objectives (Shah & Kumar, 2021). There is also a family of approaches that model videos through inferring latent actions from states, and use environment-specific action-labelled data to map these latent actions to real actions (Ye et al., 2024; Edwards et al., 2019; Schmidt & Jiang, 2024; Bruce et al., 2024). Bhateja et al. (2023) propose V-PTR which is particularly similar to our approach but only utilizes the trained ICVF encoders as a pre-trained representation for offline RL. Our method aims to use a *distance-function* rather than a pre-trained encoder to directly guide a goal-conditioned *online* RL agent. This resembles temporal distance learning methods (Pong et al., 2020; Mezghani et al., 2023) such as Dynamical Distance Learning (DDL) (Hartikainen et al., 2020) where policy-conditioned distance learning and online RL for distance minimization is alternated. However, DDL uses distances for unsupervised skill discovery and preference-learning, and importantly do not extend to internet-scale interaction data.

The most similar approach is Value-Implicit Pretraining (VIP) (Ma et al., 2023) whereby a internet-scale video-trained representation function induces a distance to shape the reward. Our method differs from VIP in a few different ways. First, our method explicitly uses temporal-difference learning as opposed to time-contrastive learning, as done in VIP. Second, we explicitly focus on downstream online RL rather than direct imitation or smooth trajectory optimization. Third, we present a bi-level pre-training procedure to not only take advantage of task-agnostic human video, but also environmental interaction data. We therefore identify with other offline-to-online methods (Xie et al., 2022; Lee et al., 2021; Agarwal et al., 2022; Zheng et al., 2023; Andrychowicz et al., 2018; Li et al., 2023) whereas VIP compares to other pre-trained representation distances. These offline-to-online methods often assume action access though which limits the scope of usable data. Our method's access to environmental interaction data dictates comparison to RLPD (Ball et al., 2023), a method which runs online RL and mixes training batches with offline data samples, as well as JSRL (Uchendu et al., 2023), a method which condenses offline data into a policy to assist online exploration.

## 3 PRELIMINARIES

Let $\mathcal{S}$ be the state space and $\mathcal{A}$ be the action space. We consider a sparse-reward Markov Decision Process (MDP), $\mathcal{M}$ defined by a tuple $(\mathcal{S}, \mathcal{A}, P, r, \gamma)$ where $P(s'|s, a)$ is the transition dynamics and $\gamma$ is the discount factor. We additionally consider a goal specified by a goal state set $\mathcal{G}$. The reward $r(s)$ is the set inclusion indicator $r(s) = \mathbb{1}[s \in \mathcal{G}]$. The objective in this setting is learn a policy $\pi$ that maximizes the expected return $\mathbb{E}_{a \sim \pi(s_t), s_{t+1} \sim P(.|s_t, a)}[\sum_{t=0}^{\infty} \gamma^t r(s_t)]$ where the expectation is taken over the policy $\pi$ and the environment dynamics.

For our experiments, we assume access to a video dataset of human egocentric interactions, $\mathcal{D}_{video}$, and a dataset of environment-specific interaction, $\mathcal{D}_{env}$. $\mathcal{D}_{video}$ contains data out of the desired domain and does not use the same embodiment as used for the target MDP $\mathcal{M}$. $\mathcal{D}_{env}$ is environment-specific data that contains actions, uses the embodiment of interest, but is either agnostic to the actual task at hand, or does not contain any successful trajectories due to the expensive nature of positive data trajectories.

## 4 VIVA : VIDEO-TRAINED VALUE FUNCTIONS

Our proposed solution for the sparse online RL case when faced with a lack of demonstrations is to develop a prior that guides the agent towards a valid goal, $g \in \mathcal{G}$. We elect to learn a value function $V(s, g)$ to give the value of any given state, $s$, in the context of the task of reaching the state $g$ optimally. As detailed in 4.1, we can train this value function to directly represent the temporal

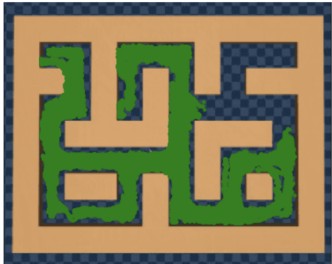 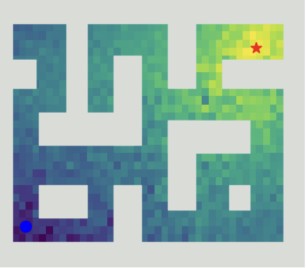 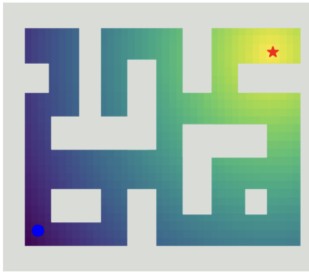

Figure 2: **Left:** A visualization of trajectories from the corrupted dataset shown in green. **Middle:** The learned ICVF values across all states with the goal at the red star. **Right:** The optimal dense reward (i.e. L2 distance) for all states with the goal at the red star.

distance from $s$ to $g$, thus giving a simple reward penalty. This allows us to create a guided reward which has an injected prior towards the goal of choice.

$$\hat{r}(s,a) = r(s,a) + V(s,g) \tag{1}$$

## 4.1 Value-function Guidance

Our desired function is $V(s,g)$, which generally yields a higher value for states closer to $g$ on the optimal path from $s$ to $g$. Since we aim to learn this model from action and reward free video data, we elect to model an Intent-conditioned Value Function, $ICVF(s,s^+,g)$, which is fully trainable from this passive data. The ICVF models the unnormalized likelihood of reaching some outcome state, $s^+$, when starting in state $s$ and acting optimally to reach some goal state $g$, otherwise known as the "intent". To precisely define the ICVF, we denote $r_g : s \mapsto r$ as a reward function corresponding to reaching any goal state. The optimal policy, $\pi_{r_g}^*$, induces a state-transition which can define the value function based on the following expectation:

$$
\begin{aligned}
P_g(s_{t+1}|s_t) &= P^{\pi_{r_g}^*}(s_{t+1}|s_t) \\
r_g(s) &= \mathbb{1}[s=g] - 1 \\
ICVF(s,s^+,g) &= \mathbb{E}_{s_0=s,s_{t+1}\sim P_g(.|s_t)} \sum_{t=0}^{\infty} \gamma^t r_{s^+}(s_t).
\end{aligned}
\tag{2}
$$

By applying a scalar shift of -1 to our sparse reward, the reward-to-go is equivalent to the negative discounted number of timesteps to reach the goal. This negative temporal distance is well-suited to be used as an additive reward penalty. Furthermore, if we use $g$ as not only the goal, but also the outcome, $s^+$, we can model the negated time to reach $g$ from $s$ if the agent were to act optimally towards $g$ thereafter. This is what we are looking for and can let us define our desired value function:

$$
\begin{aligned}
V(s,g) = ICVF(s,g,g) &= \mathbb{E}_{s_0=s,s_{t+1}\sim P_g(.|s_t)} \sum_{t=0}^{\infty} \gamma^t r_g(s_t) \\
\hat{r}(s,a) &= r(s,a) + ICVF(s,g,g).
\end{aligned}
\tag{3}
$$

We incorporate $\hat{r}$, our guided reward, into the online RL system. This allows the agent to apply knowledge of state-goal relationships contained in the learned ICVF. We note that usage of a potential-based instrinsic reward could be used for provable policy invariance as shown by Ng et al. (1999), but we observe higher variance returns which could destabilize training shown in Appendix A.3.

## 4.2 Value-Function Training

We model the ICVF as a monolithic neural network, $V_\theta(s,s^+,g)$. This differs from the original multilinear formulation, $\phi_\theta(s)^T T_\theta(g)\psi_\theta(s^+)$, since we found a monolithic architecture to produce higher-quality value functions as shown in Figure 11 in the Appendix. When working with image states, we elect to feed in learnable latent representations of the inputs to the value function. We detail the training procedure below.

Given a video dataset of image sequences, $\mathcal{D}$, we first sample a starting frame and neighboring frame $(s,s')$ from the same trajectory. Second, we sample some outcome $s^+$ from the future of the same

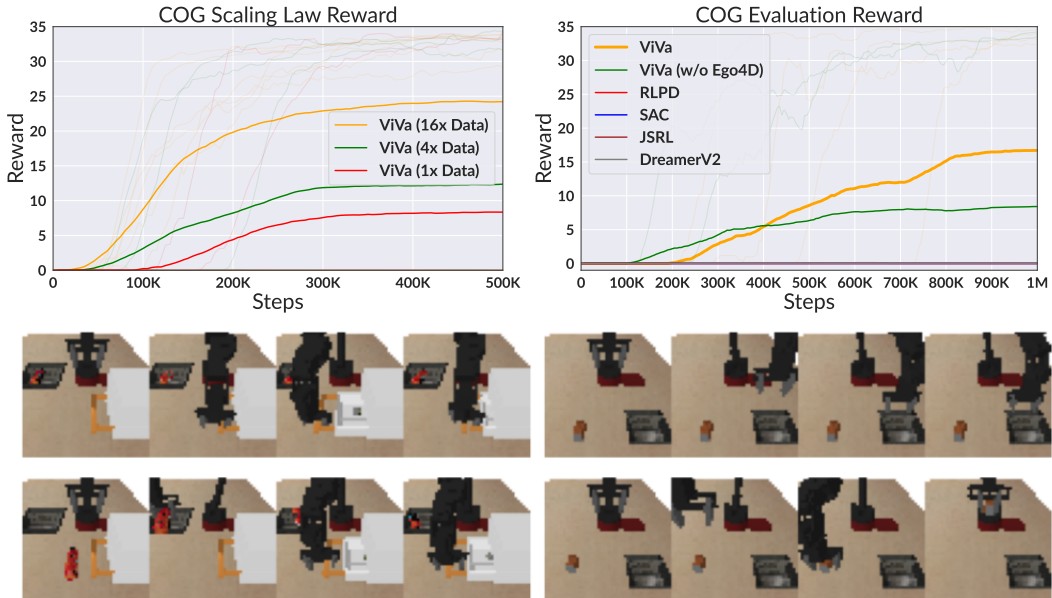

Figure 3: All plots detail the mean evaluation return computed over 10 evaluation episodes. **Left:** Online RL for pick-and-place on COG as we scale to more and more on-task data. The rows below show example off-task successful trajectories with the WidowX robot from the `drawer_prior` and `blocked_drawer` datasets. **Right:** Online RL for pick-and-place on COG when including Ego4D pretraining and off-task data sources. The rows below are a failure and a success from the `prior` dataset.

trajectory, and we sample a goal, $g$, in an identical way to $s^+$. We additionally follow Ghosh et al. (2023) in sometimes sampling identical images or random images for $s^+$ and $g$ for better training. After retrieving a sample, we minimize the temporal-difference (TD) error, in Equation 4. Inspired by Kostrikov et al. (2021a), we use the expectile regression framework with an advantage heuristic, shown in Equation 5, to relax any maximization operators. This expectile biases the objective to more strongly weight samples $(s, s')$ that are approaching $g$ under our current model of value.

$$\min_\theta |\alpha - \mathbb{1}(A \le 0)| * (V_\theta(s, s^+, g) - \mathbb{1}(s = s^+) - \gamma V_\theta(s', s^+, g))^2 \qquad (4)$$

$$A = \mathbb{1}(s = g) + \gamma V_\theta(s', g, g) - V_\theta(s, g, g) \qquad (5)$$

Essentially, if transitioning to $s'$ while conditioned on $g$ is advantageous under our current value estimates, we assume that the transition is implicitly running the optimal action to reach $g$. This allows us to update our value function without a maximum operation across actions. As a result, we just minimize the one-step TD error which is equivalent to regressing our value estimate of $V_\theta(s, s^+, g)$ towards $\mathbb{1}(s = s^+) + \gamma V_\theta(s', s^+, g)$. We use the expectile, $\alpha$, to decide how hard or soft this assumption is, with $\alpha = 0.5$ equating all samples to be equal weight, and $\alpha = 1$ forcing only using positive advantage samples for updates. As shown by Kostrikov et al. (2021a), this converges in the limit as $\alpha$ approaches 1

### 4.3 SYSTEM OVERVIEW

**Video pre-training** Using the training process described in 4.2, we first train an ICVF on Ego4D, or $\mathcal{D}_{video}$. Ego4D is a dataset of first-person camera video from hundreds of participants across many diverse scenes. This video data contains humans doing daily-life activities such as laundry, lawn-mowing, sports, gardening, and more. Approximately 3000 hours of video data is included and we reshape to $128 \times 128$ and apply a random crop augmentation further detailed in Appendix A.1. As detailed earlier, we utilize a -1 reward shift for the self-supervised reward targets to ensure the

value to-go matches a temporal distance as desired. We elect to sample future outcomes and goals from the same trajectory 80% of the time and use a 10% chance for both choosing random goals or goals equal to current sampled state. Lastly, we choose an expectile of 0.9 which ensures backups are biased to occur stronger for transitions where the advantage heuristic is positive. This expectile allows for the convergence guarantees in optimal value function learning as the expectile approaches 1 shown in Kostrikov et al. (2021a). We utilize ResNetv2 (He et al., 2016) on JAX (Bradbury et al., 2018) as our neural architecture and functional paradigm for this video pre-training. We encode the three input images, $(s, s^+, g)$ with the ResNet before passing them into an ensemble of two 2-layer MLPs for min-Q learning.

**Environment fine-tuning** Secondly, we use available environment data, $\mathcal{D}_{env}$, to finetune the ICVF. The finetuning is done exactly the same way as pre-training but with environment video data. This finetuning brings the model into the domain of the RL task and can help to develop setting-specific features relevant to tasks in the environment. We hypothesize usage of $\mathcal{D}_{video}$ will develop general visual features and fusion between the input and goal images. Furthermore, it can learn priors about the cause-and-effect of manipulation. Alternatively, the fine-tuning on $\mathcal{D}_{env}$ will help to develop task-specific features and the visual dynamics of the target environment.

**Guided online RL** After our value function is trained, we lastly run online RL and utilize the available environmental data, $\mathcal{D}_{env}$, as prior

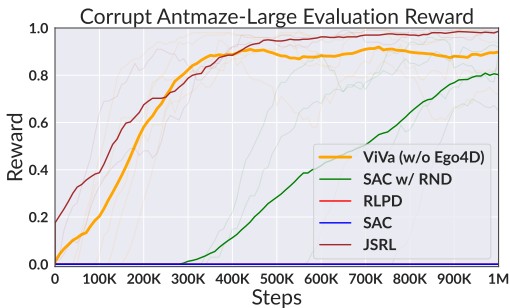

Figure 4: The online evaluation return in AntMaze when training ViVa with corrupted data. As seen, learning a value-function prior for online RL provides a more generalizable reward model when offline rewarded data is absent. Learning a behavioral prior also works in this setting.

data. Specifically, every batch update for online RL includes 50% sampled online data from the replay buffer and 50% sampled offline data from the prior dataset. The addition of the prior data into our RL training assists online exploration by providing offline trajectories to backup across and explore which may not be otherwise explored online. We run experiments with this system set up to study and analyze generalization characteristics, performance, and data scaling properties. We provide details of our chosen algorithms Soft Actor-Critic (Haarnoja et al., 2018b) and its DrQ variant (Kostrikov et al., 2021b) in the Appendix. As shown by Haarnoja et al. (2018b), soft policy iteration is shown to converge.

## 5 RESULTS

We analyze ViVa through different lenses to understand the benefits of video-trained value functions for downstream online RL. Specifically, we seek to study the **generalization** capabilities of ViVa in providing effective guidance for tasks it has not been provided data for, the **performance** of ViVa in difficult control tasks, and the **scaling** properties of ViVa as more diverse data is incorporated in greater quantities.

### 5.1 BASELINES

We also choose to compare to other methods which take advantage of offline data to determine whether **video-trained value functions** are an effective mathematical object for representing a prior for online RL. We firstly compare against Reinforcement Learning with Prior Data **(RLPD)**, a method which simply includes offline prior data in the update batches exactly as we do in ViVa . Importantly, RLPD only uses extrinsic reward signals and does not pretrain or finetune a value for relabeling offline data as ViVa does. We also compare against Jump-start Reinforcement Learning **(JSRL)** which learns a behavioral prior policy from offline data and then runs online RL by executing the learned prior policy for $N$ random steps and then giving control to the agent's policy until termination. This method aims to condense prior experience into a policy for improving exploration towards desired goals. For our experiments, we train an imitative policy from offline data and use that as the behavioral prior for JSRL. Lastly, we use vanilla **DreamerV2**, a competitive world-model

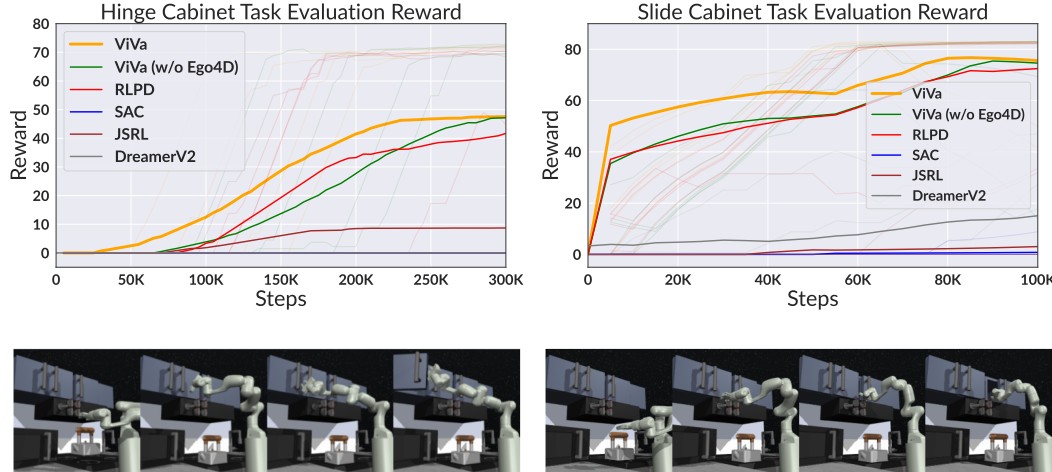

Figure 5: All plots detail the mean evaluation return computed over 10 evaluation episodes. **Left:** Online RL for the Hinge Cabinet task in FrankaKitchen. The bottom row is an image trajectory of a demonstration of opening the hinge cabinet. **Right:** Online RL for the Sliding Cabinet task in FrankaKitchen. The bottom row is an image trajectory of a demonstration of opening the sliding cabinet.

approach for online RL (Hafner et al., 2022) that uses latent imagination of rollouts for training. We use these baselines to study the importance of explicitly learning a value function from prior data as opposed to directly including it in the replay buffer or learning a behavioral prior from it. We also compare against vanilla Soft-Actor Critic (**SAC**) and ablate Ego4D pre-training from ViVa to further analyze our method.

## 5.2 EXPERIMENTS

**Corrupted AntMaze** We first use the D4RL AntMaze (Fu et al., 2021) environment to visually analyze the robustness to states seen outside of the training distribution. Environment and training details are further expanded upon in Appendix A.3. We modify the D4RL `diverse` prior dataset which includes the training transitions of a random start goal-reaching policy. Importantly, we corrupt this dataset by removing all trajectories containing points near the goal-region as shown in Figure 2. We train a 3-layer Multilayer Perceptron with 512 units each using Equation 4 as the training objective and display the learned value function, after 45 minutes of training on a Tesla V100 16GB GPU, in Figure 2. Evidently, we observe generalization to the goal region when it has not been seen during value training. The benefits of this generalization can be seen when running downstream online RL are shown in Figure 4. We conclude that learning a simple ICVF network on offline data is enough to develop a prior that generalizes to the unseen goal and prevents the failures that RLPD exhibits on sparse-reward tasks when the offline dataset doesn't contain the goal. Our comparison shows that JSRL exhibits similar extrapolation to new goals in the space of expert actions as opposed to values. However, this similar extrapolation ability seems to fail when introduced to more complex visual environments. In these settings ViVa is able to take advantage of Ego4D pre-training whereas JSRL cannot.

**RoboVerse off-task transfer** We use the RoboVerse (Singh et al., 2020) simulator (COG) to test whether ViVa can generalize to new tasks never seen before in a visual domain, rather than state-based. This simulator has a variety of settings and accompanying datasets using a 6 DoF WidowX-250 robot on a desk. We choose to evaluate on a pick-and-place task to move a randomly placed object into a tray – this task has two datasets of interest, one of 10K grasping attempts (with around 40% success), known as the `prior` dataset, and one `task-specific` dataset of 5K placing attempts (with around 90% success) labeled with rewards. For our experimental setup, we choose to exclude the `task-specific` dataset to emphasize the absence of positive demonstration data. During training, we combine the `prior` data with various other off-task sets which contain inter-

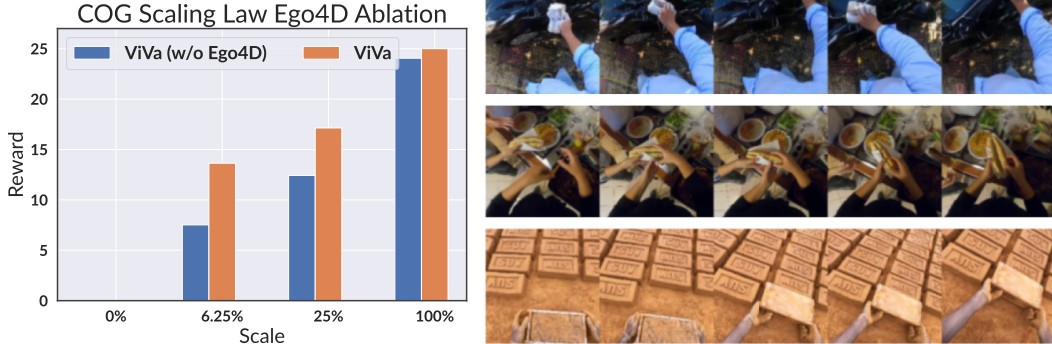

Figure 6: **Left:** An ablation study of including Ego4D pretraining or not across different environment finetuning data availability levels. At 0%, we are evaluating a random value function and the zero-shot performance of an Ego4D trained value function. **Right:** Each row is a randomly sampled trajectory from the Ego4D training showing car washing, cooking, and construction (from top to bottom).

actions with an open drawer, a closed drawer, and an obstructed drawer. All these datasets contain no rewards as they do not match the desired reward we are using. We train ViVa for 9 hours on a v4 TPU with these datasets and use it to guide online RL for pick-and-place. We leave details of the training and environment in the Appendix. The results in Figure 3 show that ViVa succeeds in solving the task whereas plainly sampling the available data offline fails. This is since the offline data includes no rewards so RLPD fails to benefit from offline batch updates. On the other hand, the imitative prior that JSRL uses does not explore the right areas which slows down learning. Interestingly, this experiment shows that ViVa is able to take advantage of diverse off-task environmental and video data to inform goal-reaching. To concretize this conclusion, we ablate these data sources to show that this is what enables guidance to unseen goals in the Figure 9 in Appendix A.4.

To more deeply understand how the trained ICVF behaves on out-of-distribution examples, we also plot value curves over trajectories of failure demonstrations and unseen successes in Figure 7. As shown, ViVa provides guidance towards unseen goals, resembling how a control value function trained on positive demonstrations does. Similar to the AntMaze experiment, ViVa provides generalization to unseen goals and assists downstream online RL while also taking advantage of Internet-scale video data. We analyze the usage of how ViVa behaves as more data is available and how useful this Internet-scale pre-training is in the experiments below.

**RoboVerse scaling law** In this experiment, we seek to assess whether a greater amount of task-relevant data has a positive effect on the downstream RL guidance. We train ViVa on a varying amount of data (from the `task-specific` and `prior datasets`) for 5 hours on a v4 TPU and then examine the online performance. We elect to include only the `prior` dataset in the online RL phase since including the `task-specific` data would significantly simplify the problem. As shown in Figure 3, we can see there is a strong performance increase as data scales upwards. This shows that ViVa benefits from the diversity and coverage of its training data and has positive scaling behavior.

**RoboVerse pre-training ablation** We run the same analysis in our scaling law, but we remove environment-agnostic, task-agnostic video to assess the direct impact of Ego4D pretraining. In Figure 6 we can see a diminishing yet positive return from pre-training the value function on internet-scale Ego4D video. Specifically, in the low-data regime, we observe a 2x increase in performance when including Internet-scale video. This demonstrates the effective transfer to online RL from including videos of interaction data supporting our initial hypothesis of developing a goal-reaching prior for guidance. Although, the poor zero-shot performance depicts the importance of fine-tuning on environmental data given the significant domain shift from Ego4D to RoboVerse.

**Franka Kitchen** Lastly, to evaluate on a more difficult robotic benchmark, we run ViVa on the FrankaKitchen (Fu et al., 2021; Gupta et al., 2019) environment which simulates a 9-DoF Franka robot tasked to interact with different objects in a kitchen. We use datasets of $\sim$ 1K failure trajectories when attempting to interact with the Hinge Cabinet and Sliding Cabinet as the environmental

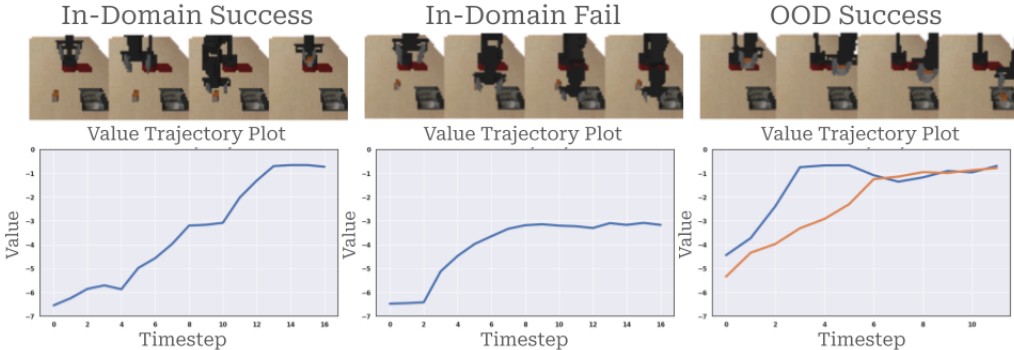

Figure 7: The top shows the image trajectory being evaluated and the bottom is the corresponding value function plot. **Left:** Value across a successful trajectory conditioned on a picking goal. **Middle:** Value across a failure trajectory conditioned on a picking goal. **Right:** Value on an unseen placing trajectory with an unseen placing goal. The blue is our model generalizing, and the orange reference is an optimal value function learned on the placing task.

interaction data to finetune the video-pretrained value function. Finetuning is run for 2.5 hours on a v4 TPU. Results of each task with and without video pretraining are shown in Figure 5. We observe that RLPD works well due to the negative reward shift that encourages the agent to be near the terminal states of the prior data. JSRL works poorly yet still succeeds on some seeds since imitating the interaction failures allow for exploring near the right area. Evidently, the inclusion of Ego4D pretraining tends to improve sample-efficiency.

# 6    DISCUSSION

In this paper, we proposed a method for transferring goal-reaching priors found in video data to downstream online RL problems by learning an intent-conditioned value function. This method can enable sparse-reward task solving, generalization to new goals, and positive transfer between tasks. Our analysis of ViVa illustrates the importance of using value function pre-training on video data ~~as opposed to other methods of utilizing prior data~~. Our scaling experiments show that this is due to the wide support that this method can take advantage of, namely from the availability and generality of video data as well as the lack of assumptions for value learning.

Comparisons with JSRL depict the superiority of value functions as a representation of prior data as opposed to classical imitative policies. We hypothesize this is because value learning uses a method akin to shortest-path finding within data to discover an underlying temporal structure as opposed to naively matching the next action. Furthermore, direct imitative policies prevent support from action-free data sources such as Ego4D. However, latent-imitation methods could be explored to leverage actionless datasets. Regardless, the ViVa paradigm should therefore provide insight to RL practitioners looking to harness extra data and ameliorate the absence of rewarded prior data.

**Limitations and future work** We note that a limitation of value functions is the weak zero-shot extrapolation ability when far out of domain. This can be seen through the poor 0% scale performance shown in Figure 6 which is presumably because RoboVerse is significantly different than Ego4D. But when there is fine-tuning involved (shown at scales larger than 0% in Figure 6), this Ego4D pretraining helps, offering 2x performance boost in the low-data regime. These results make it clear that pre-training offers a way to make fine-tuning more effective, but cannot work on its own as it'd need task-relevant data. A direction of future work would be to find ways to encode more explicit forms of abstraction in the value function in order to extrapolate deeply when given only off-domain pre-training data (such as Ego4D). This would help to improve pure zero-shot performance when given no environment data.

We also notice that ViVa utilizes some action-labelled robotic data for fine-tuning which is assumed to be exploratory or somewhat relevant to the downstream task. An exciting future direction would be to pair ViVa without fine-tuning with an exploration algorithm online to run the fine-tuning during the online RL phase, thus simplifying the training pipeline by removing a separate fine-tuning phase. This method would also allow for resolving value errors through collecting counterfactual

examples since these errors can be detrimental to performance by forcing the agent into states that are erroneously near the goal. This way, ViVa could even be used to form a curriculum based on state-values or uncertainties in state-values to guide exploration in harder problems. Lastly, a natural extension includes utilizing language goals for the intent-conditioned value function, harnessing multi-modal features, and extending into the real-world.

## 7 REPRODUCIBILITY STATEMENT

We include important training parameters in our system overview, in Section 4.3. These include image shapes, augmentation choices, reward shift, and hyperparameters that control the data sampling for training. In Section 5, we include domain specific parameters as well as the datasets used for fine-tuning ViVa . Lastly, we include an Appendix with fine-grained training details, datasets, and code-bases used. The Appendix expands upon details mentioned in the main paper and gives parameters for exactly reproducing the models we have trained on the code repositories we used.

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

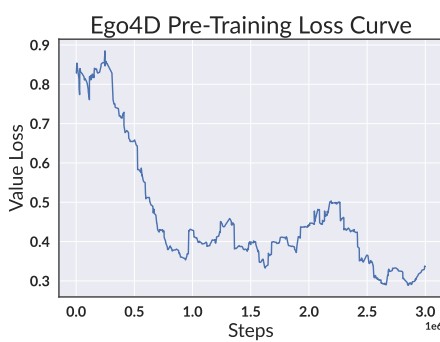

Figure 8: ICVF Ego4D value loss over training.

| Hyperparameter | Value |
|---|---|
| $p_{randomgoal}$ | 0.1 |
| $p_{trajgoal}$ | 0.8 |
| $p_{currgoal}$ | 0.1 |
| reward_scale | 1.0 |
| reward_shift | -1.0 |
| $p_{samegoal}$ | 0.5 |
| intent_sametraj | True |
| Encoder | ResNet-v2 |
| MLP Hidden Dims | [256, 256] |
| Value Ensemble Size | 2 |
| Optimizer Learning Rate | 6e-5 |
| Optimizer Epsilon | 0.00015 |
| Discount | 0.98 |
| Expectile | 0.9 |
| Target Update Rate | 0.005 |
| Batch Size | 64 |

Table 1: ICVF Ego4D Training Settings. We include parameters from the ICVF public code base to control the image sampling mechanism.

# A APPENDIX

## A.1 VIVA TRAINING

We pre-train ViVa on the Ego4D video dataset. We use the public ICVF codebase and use settings shown in Table 1. We preprocess the video dataset by shaping to $256 \times 256$, center cropping the middle $224 \times 224$, then resizing it to $128 \times 128$. The ICVF itself is structured with an encoder which converts the state, future outcome, and goal into embeddings. For the encoder, we utilize the 26-layer ResNet-v2. The training loss is displayed in Figure 8. We train with 1 v4 TPU for 1.5 days.

Once embedded, we concatenate the latents and pass them into an ensemble of 2 Multilayer Perceptrons, each with LayerNorm and to produce the value estimate. We train the ICVF for 1 million steps. For RoboVerse and Franka Kitchen, we apply the same exact training process, but on a the fine-tuning dataset. Antmaze doesn't utilize pretraining and functions on states, so it has a different setup. For our final experiments, we swept across checkpoints to identify strong value functions to run online RL with.

## A.2 ONLINE RL

When running online RL with a trained ICVF, we formulate our reward as:

$$\tilde{r}(s,a) = r(s,a) + ICVF(s,g,g) \qquad (6)$$

However, we experimented with a different approach where

$$\tilde{r}(s,a,s') = r(s,a) + (\gamma\Phi_g(s') - \Phi_g(s)) \qquad (7)$$

$$\Phi_g(s) = ICVF(s,g,g) \qquad (8)$$

which follows the potential-based reward shaping strategy as formulated by Ng et al. (1999). They show a learned Q-function under the proposed reward transformation is:

$$\tilde{Q}_g^*(s,a) = Q_g^*(s,a) - \Phi_g(s) \qquad (9)$$

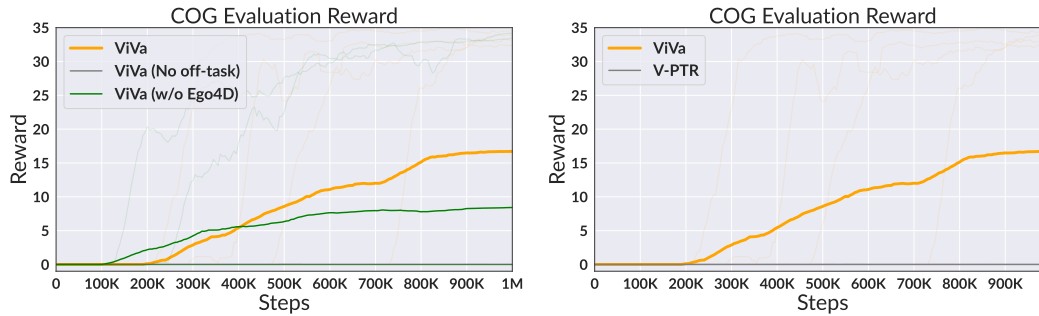

Figure 9: **Left:** An experiment ablating out off-task data and Ego-4D pre-training. As seen, off-task data and off-environment pre-training are sigificant for performance boost. **Right:** Comparison between V-PTR and ViVa on the COG pick-and-place task showing how ViVa's design of reward guidance trumps simple representation transfer on COG.

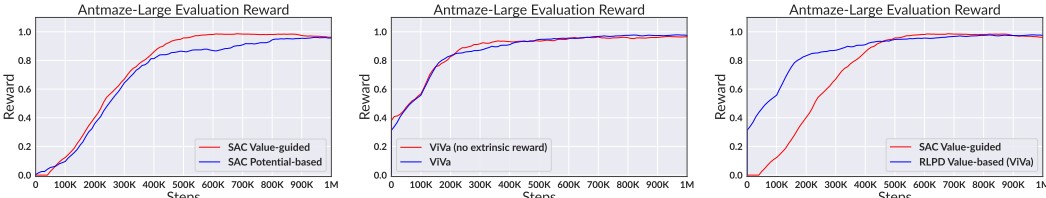

Figure 10: **Left:** Comparison in AntMaze between using value potential or pure value as the reward augmentation. **Middle:** Comparison in AntMaze between agents given access to extrinsic reward labels or not. **Right:** Comparison between agent given prior data access (RLPD) and agents given 0 prior data (SAC)

which is evidently invariant of actions and thus admits the same optimal policy. Although this is favorable in theory, in practice we observed no changes in results except variance in policy rollout returns which could destabilize training. This was tested in Antmaze by training an ICVF on the full Antmaze dataset and utilizing the potential-based shaping reward versus the simple value-guided reward as shown in Figure 10.

## A.3 ANTMAZE

**Value training** Our first experiment involves the AntMaze environment specified in the D4RL experiment suite. It is build upon Mujoco and controls an 8 DoF ant with 4 legs through a maze. It starts in the bottom left and is tasked to reach the top right using a sparse reward. In practice, we do not utilize any ICVF Ego4D pretraining since we run this experiment in a state-based fashion. The state is 29-dimensional including positions, velocities, angles, and angular velocities. We use a different ICVF setup for training too. Specifically, we utilize a discount of 0.999, a learning rate of 3e-4 and an epsilon of 1e-8. We use a 3 layer, 512 unit MLP with LayerNorm as the value function. We experimented with using the original multilinear formulation proposed by Ghosh et al. (2023) but noticed early collapse during training as well as noisy values, shown in Figure 11. This motivated our choice to use a single, monolithic neural architecture to represent value.

**RL training** We run on the RLPD public codebase and detail RLPD hyperparameters in Table 2. RLPD simply runs the Soft Actor-Critic algorithm but adds offline sampling and some extra design choices as detailed in their paper. We edit every update batch reward by adding the ICVF value for the current state conditioned on the goal times 0.001. We use 5 seeds for all baseline experiments.

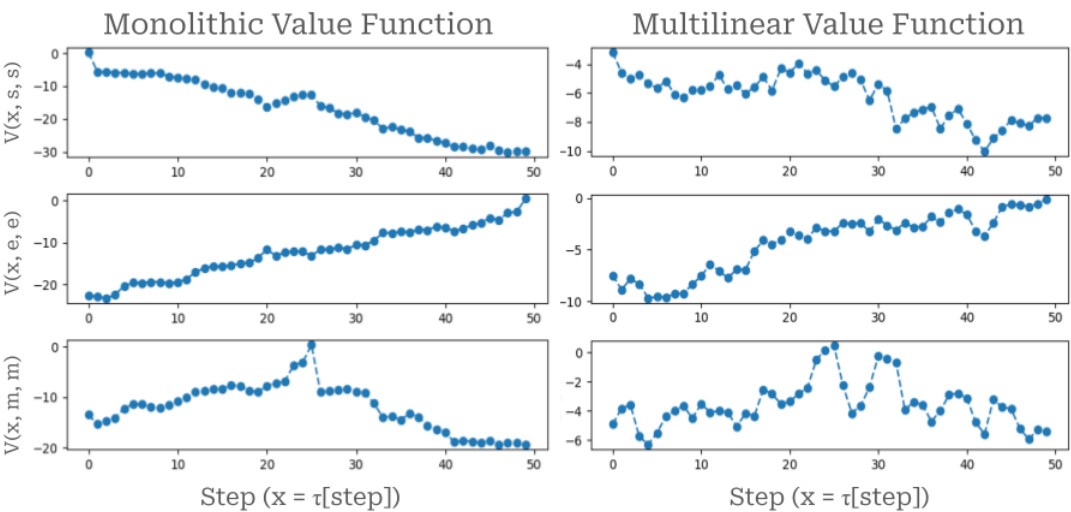

Figure 11: These plots show trained value function curves across time on a trajectory from s to m to e, representing start, middle, and end, respectively. We denote x as representing the state in the trajectory at a given timestep. Each row is a comparison of values when setting the goal-conditioning to start, middle, or end. As depicted, the monolithic values much more smoothly express distance from the start, middle, or end as we move across the trajectory.

| Hyperparameter | Value |
|---|---|
| CNN Features | (32, 64, 128, 256) |
| CNN Filters | (3, 3, 3, 3) |
| CNN Strides | (2, 2, 2, 2) |
| CNN Padding | "VALID" |
| CNN Latent Dimension | 50 |
| Update-to-Data Ratio | 1 |
| Offline Ratio | 0.5 |
| Start Training | 5000 |
| Backup Entropy | True |
| Hidden Dims | (256, 256) |
| Batch Size | 256 |
| Q Ensemble Size | 2 |
| Temperature LR | 3e-4 |
| Init Temperature | 0.1 |
| Actor LR | 3e-4 |
| Critic LR | 3e-4 |
| Discount | 0.99 |
| Tau | 0.005 |
| Critic Layer Norm | True |
| Horizon | 40 |

Figure 12: RLPD Settings for COG RoboVerse and FrankaKitchen

| Hyperparameter | Value |
|---|---|
| Update-to-Data Ratio | 20 |
| Offline Ratio | 0.5 |
| Start Training | 5000 |
| Backup Entropy | False |
| Hidden Dims | (256, 256, 256) |
| Q Ensemble Size | 1 |
| Temperature LR | 3e-4 |
| Init Temperature | 1.0 |
| Actor LR | 3e-4 |
| Critic LR | 3e-4 |
| Discount | 0.99 |
| Tau | 0.005 |
| Critic Layer Norm | True |
| Horizon | 1000 |

Figure 13: RLPD Settings for Antmaze

| Method | AntMaze Corrupt | COG Pick-Place | Franka Hinge | Franka Slide |
|---|---|---|---|---|
| ViVa | N/A | **8.73 ± 13.48** | **30.38 ± 32.07** | **62.64 ± 12.85** |
| ViVa (No Ego4D) | 0.89 ± 0.14 | 6.42 ± 11.25 | 14.91 ± 25.20 | 54.75 ± 14.98 |
| JSRL | **0.95 ± 0.04** | 0 ± 0 | 0 ± 0 | 1.68 ± 3.76 |
| DreamerV2 | N/A | 0 ± 0 | 0 ± 0 | 15.11 ± 22.36 |
| RLPD | 0 ± 0 | 0 ± 0 | 21.06 ± 23.64 | 54.38 ± 19.71 |
| SAC | 0 ± 0 | 0.01 ± 0.03 | 0 ± 0 | 0.44 ± 0.98 |
| ViVa | N/A | **16.71 ± 16.73** | **47.56 ± 33.64** | **75.61 ± 11.46** |
| ViVa (No Ego4D) | 0.9 ± 0.13 | 8.42 ± 14.58 | 47.15 ± 33.36 | 74.61 ± 15.06 |
| JSRL | **0.98 ± 0.01** | 0 ± 0 | 8.7 ± 23.00 | 3.06 ± 6.84 |
| DreamerV2 | N/A | 0 ± 0 | 0 ± 0 | 25.32 ± 30.71 |
| RLPD | 0 ± 0 | 0 ± 0 | 41.704 ± 33.40 | 72.47 ± 21.44 |
| SAC | 0 ± 0 | 0.02 ± 0.03 | 0 ± 0 | 0.80 ± 1.79 |

Table 2: Experimental Suite Results. The top set of results are at the halfway point through the online RL training process. The bottom rows are metrics at the final step.

## A.4 COG RoboVerse

We use the RoboVerse simulator, publicly located here, which simulates a WidowX robot through PyBullet. We use the datasets created in the COG paper which is publicly located here. We run experiments on the pick-and-place task which sparsely rewards the agent for picking up a target object randomly placed on a table and placing it into a silver tray. For ICVF fine-tuning, we utilize a number of data combinations for different experiments detailed in the paper, but select from the main group of COG datasets: `pickplace_prior`, `pickplace_task`, `DrawerOpenGrasp`, `drawer_task`, `closed_drawer_prior`, `blocked_drawer_1_prior`, and `blocked_drawer_2_prior`. We only include `pickplace_task` in the scaling law and elect to remove it for all other experiments.

During the online RL phase, we adopt the same RLPD system but we use the DrQ regularization methods for image-based RL. Specifically, we utilize the D4PG (Barth-Maron et al., 2018) visual encoder. We attach experimental hyperparameters in Table 12 and use 8 seeds each. We additionally compare our method to V-PTR, a similar method which uses the trained representations from the ICVF rather than the actual value network outputs. V-PTR uses the trained encoders to map the observations into an embedding space for the policy network to learn on. Since our method uses the notion of distance itself and more actively enforces this signal directly into the reward, we hypothesize it'd be more directly useful for sparse reward RL. Our comparison results are in Figure 9 which motivate our decision to use values directly. We additionally ablate off-task data and Ego4D pre-training to show the effect of each data source in Figure 9.

## A.5 FRANKA KITCHEN

Our final experiment uses the Franka Kitchen environment available on D4RL here which simulates a 9-DoF Franka Robot in a kitchen environment. We control the robot in joint velocity mode clipped between -1 and 1 rad/s. The 9 degrees of freedom are 7 joints and 2 fingers of the gripper. We analyze two tasks which are opening the sliding cabinet and opening the hinge cabinet. These tasks are specified with a sparse reward. As mentioned in the paper, the datasets we used contain failed interactions with the target objects. We collect this data by controlling the robot with expert demonstration actions with added Gaussian noise. We then filter out all successes from this data to form our dataset. The hinge failures dataset contains 1013 trajectories, whereas the sliding door dataset contains 630 trajectories. These trajectories are 50 steps each. The RLPD settings for FrankaKitchen are the exact same as for RoboVerse but we use a horizon of 50 steps rather than 40, and we run 6 seeds per baseline.

