# OpenReview forum: "ViVa: Video-Trained Value Functions for Guiding Online RL from Diverse Data"
_ICLR.cc/2025/Conference — Submitted to ICLR 2025_

### Official Review · Reviewer_U8sC · 2024-10-18

**Soundness:** 2
**Presentation:** 2
**Contribution:** 2
**Rating:** 3
**Confidence:** 4

**Summary:**

This work proposes ViVa to learn prior knowledge from large-scale video datasets and then fine-tune on in-domain data. It encodes the video to representations first and then uses shallow MLP layers to output the value function.

**Strengths:**

* An ablation study is conducted.
* Deep studies help investigate the succeed of the method on tasks (Fig. 2 and 7).

**Weaknesses:**

* The experiments are all in simulators, not in the real-world. It is unclear how the ViVa works on real-image tasks.
* Mathematical formulation is unclear (see questions).
* In corrupted AntMaze, JSRL is better than ViVa. Why is that, and what is the point of conducting the experiment then?
* In the setting of demonstrations with rewards, the author should compare it with more baselines, such as Adril. In the setting of demonstrations without rewards, the author should compare against methods that learn a world model like Dreamer.

# Minor:
* incorrect use of ” ”,  use “” (line 43)
* no space before whereas (line 129)
* when COG first appears, it is not defined.
* “Figure” 5 in line 397.
* start appendix with section A.

**Questions:**

* See weakenss1
* How do we intuitively understand why V-PTR is less performant than ViVa?
* What is f in line 207?
* Is equation 4 minimizing over V theta? And the second line is st.t?
* What is the difference between RLPD and ViVa in Antmaze? They all only learn from the demonstrations.
* isn’t the marginal increase in performance in Figure 6 indicating that there is no need for heavy pre-training? As stated in the limitation, “RoboVerse is significantly different than Ego4D”, so the pre-training phase only makes sense for relevant tasks to Ego4D. I believe further investigation is needed to look into this relevance, which again might be relevant to the gap between simulators and the real world.
* Following the previous, this work motivates the use of Internet-scale datasets to accelerate learning in the abstract, but some experimental results support the opposite view. If not using the Internet-scale datasets, I do not see how this work separates from other works on imitation learning.

---

> ### Author Response · Authors · 2024-11-23
>
> Thank you for your thoughtful review of our paper. In response to your concerns, we have updated the paper to improve descriptions of our framework and added experiments. We respectfully comment on four points:
> 1. **“marginal increase in performance ... no need for heavy pre-training?”**  We find that it yields considerable benefits in the low prior robot data setting (in Fig 6, it improves performance by 2x), is not especially expensive to acquire (1.5 days pre-training on a single TPU once and brief fine-tuning for ~3 hrs), and provides a way of incorporating broad external knowledge into downstream RL.
> 2. **“this work motivates the use of Internet-scale data..., but some experimental results support the opposite view.”** We find on Figures 3, 5, and 6 that including Internet-scale pre-training improves performance consistently, especially in the low-data regime shown in Figure 6 (6.25% scale).
> 3. **“the pre-training phase only makes sense for relevant tasks to Ego4D.”** In the Limitations section, we speak towards the weak zero-shot performance of the model (first column of Figure 6, at 0% scale). Specifically, we observe that Ego4D pretraining only helps when there’s fine-tuning involved. However, this does not mean that pre-training is not useful, as it consistently improves performance as finetuning data is introduced. We have edited the Limitations in Section 6 to highlight the inability of ViVa to assist guidance with 0% environment data as a motivation to explore creating even more general priors to potentially improve zero-shot performance.
> 4. **“If not using the Internet-scale datasets, I do not see how this work separates from other works on imitation learning.”** Without internet-scale training, this method does indeed relate to other works that do action-free reward shaping – we add mentions to significant works in Section 2 - Related Work (Edwards et al 2019). Our paper does, however, focus on the design necessary for scaling to large video datasets and non-demonstration data all while displaying positive transfer even when deeply out-of-domain.
>
> **Baselines:** We add a DreamerV2 baseline to our experimental suite. Note that AdRIL is an imitative method that requires demonstration data. Our method, on the other hand, is designed to run with arbitrary training data, not only demonstrations. For example, in COG, partial trajectories and failures are used for training, and in FrankaKitchen, we use noisy failures for training.
>
> **Corrupted AntMaze:** As you mention, since the Antmaze domain has no video-only (actionless) pre-training data, approaches like JSRL, which assumes that all the pre-training data have action labels, can be competitive with ViVa. The purpose of this experiment is not to claim that ViVa is SoTA for Antmaze, but rather that our approach outperforms others that make the ``video’’ assumption: that pre-training data may not have actions. On more complex tasks, where we can pre-train on a large action-less dataset, we see that JSRL underperforms ViVa by a significant degree (Figure 3, 5). We have updated the discussion in Section 5.2 (lines 367-8) to discuss the performance of JSRL in Antmaze.
>
> **Real-world Experiments:** This is an interesting avenue for future research, but we believe is beyond the scope of our paper. We execute on multiple simulated domains, including those with realistic-looking image environments, and run different tasks and ablation studies to conduct a thorough analysis of our method. Executing online RL on real robotic systems is a significant infrastructural / hardware challenge, noisy, and difficult to reproduce – and would require us to shift our paper from an reproducible and controllable analytical study.
>
> **Minor:**
> > Notation
>
> We remove mention of “f”. We are minimizing V over theta (and edit Eq 4) and separate out the advantage as a new line (Eq 5).
>
> > RLPD vs. ViVa
>
> This is a great question! Even though RLPD and ViVa both learn from the same data, they utilize the offline data differently. Both methods optimize the online RL objective using the prior offline data to train the online value function (and policy). ViVa (our method) additionally uses the offline data to train an ICVF, a goal-conditioned value function of sorts, which is used for online reward shaping. We explicitly stated this in line 316 on the original submission: “Importantly, RLPD only uses extrinsic reward signals and does not pre-train a value for relabelling offline data as ViVa does,” but we have added extra clarity to this statement in Section 5.1.
>
> > V-PTR vs. ViVa
>
> Good question! V-PTR uses the trained encoders from the ICVF training to define a latent space to run online RL with. Our method uses the notion of distance itself and more actively enforces this signal directly into the reward. We hypothesize this is more directly useful for sparse reward RL. We add an experiment in Section A.4 of the Appendix in Figure 9 comparing V-PTR and ViVa.

---

> > ### Comment · Reviewer_U8sC · 2024-11-24
> >
> > Thank you for the response.
> > The draft has significantly changed to add crucial explanations and discussion, but the original submission is very confusing (see questions about corrupted AntMaze and mathematical formulation). I do not think this paper is ready to be published.
> > It needs more explanation regarding the pre-trained model on real videos and being fine-tuned on simulation data. One may alternatively use a pre-trained model on a lot of simulation data to conduct the same improvement. Hence, I am not fully convinced by the experiments, and I believe more ablation studies and better communication to convey the results are needed.
> >
> > I prefer to maintain my current score.

---

> > > ### Comment · Reviewer_U8sC · 2024-11-25
> > >
> > > Another doubt, is there any reason why the authors did not run DreamerV3 but DreamerV2?

---

> > > > ### Author Response · Authors · 2024-11-28
> > > >
> > > > > Another doubt, is there any reason why the authors did not run DreamerV3 but DreamerV2?
> > > >
> > > > We used DreamerV2 because it was the simplest to setup with our external Gym environments under the limited discussion period (pip-installable). We will update the final paper with a DreamerV3 baseline.

---

> > > > > ### Comment · Reviewer_U8sC · 2024-11-28
> > > > >
> > > > > Thank you for the detailed response and summary of the updates.
> > > > >
> > > > > > pre-training on lots of simulation data
> > > > >
> > > > > This can be done by collecting a lot of demonstrations, though I understand this may be irrelevant to the topic and the core setting of the paper.
> > > > >
> > > > > > I believe more ablation studies...
> > > > >
> > > > > It is also crucial to compare the ratio of optimal data (successful examples with full observation) in the fine-tuned dataset vs. the overall performance. This can give a view of how ViVa is distinguished from other imitation learning works (as they need optimal data as stated in the author's rebuttal above). If you already did so, please correct me.
> > > > >
> > > > > > ... better communication to convey the results are needed
> > > > >
> > > > > To sum up, I think the new response addresses my concerns. I would suggest the authors rewrite the introduction section to highlight the contributions, which you already did in the rebuttal, since after re-reading the section several times, I still do not have a clear idea of the contributions.
> > > > >
> > > > > If the authors can finish the baseline of Dreamer V3 + the ablation study of the optimal ratio mentioned above + the intro section during the discussion period, I am more than happy to raise the score.
> > > > > I understand that there is not much time left and reviewers should not ask for more results, but since Dreamer (though not explicitly mention which version, I think it is straightforward to understand it as the newest version) has already been requested in the review, I think it is fair.
> > > > >
> > > > > In the case the authors cannot finish all, I will also consider raising the score as it is evident that the revised submission is much better than the original one.

---

> > > > > > ### Author Response · Authors · 2024-12-03
> > > > > >
> > > > > > Thank you for your response and understanding. We ran our experimental suite with DreamerV3 and observed that DreamerV2 performs better. We have now specified this in our paper and kept the DreamerV2 results for a better baseline comparison.
> > > > > >
> > > > > > > It is also crucial to compare the ratio of optimal data (successful examples with full observation) in the fine-tuned dataset vs. the overall performance.
> > > > > >
> > > > > > We agree with the importance of this experiment. We include this in the original paper in Figure 3. This experiment continuously scales up the amount of on-task data containing successful demonstrations of placing. We find that as on-task data increases, ViVa’s performance scales accordingly.
> > > > > >
> > > > > > > I would suggest the authors rewrite the introduction section to highlight the contributions
> > > > > >
> > > > > > We have now edited the paper’s introduction section to align more heavily with our rebuttal discussion. We cannot upload the revision now due to the edit deadline, however, we now highlight the contributions with explicit reference to the low-data regime as well as our method’s difference in assumptions from classic imitation and why that’s important.

---

> > > ### Author Response · Authors · 2024-11-28
> > >
> > > > I believe more ablation studies and better communication to convey the results are needed
> > >
> > > We would be happy to include additional ablation studies, if the reviewer had any specific comparisons in mind. The original submission contains ablations of Ego4D pre-training on all experiments, and in the first revision, we added prior sampling ablation as well as extrinsic reward ablation on AntMaze. In this revision, we have included additional ablations of off-task data for the Roboverse suite to show how the external datasets affect goal generalization. We additionally made edits to Section 5.2 to describe these experimental results in greater detail. We strongly agree with the reviewer that ablations can offer deeper insight into the experimental results and would love to hear what specific ablations the reviewer thinks would help to be “fully convinced by the experiments”.
> > >
> > > > It needs more explanation regarding the pre-trained model on real videos and being fine-tuned on simulation data.
> > >
> > > Could you explain what specific aspects of the pre-training process remain difficult to understand? During the discussion period, we improved Figure 1 to more clearly demonstrate the pre-training / finetuning process, and we further updated Section 4.3 to discuss how the pre-training, finetuning, and online RL is done. In Appendix A.1, we added many relevant Ego4D pre-training details, including data, compute, and pre-training loss curves. We would be happy to add additional information that you think is missing to improve the clarity of this section.
> > >
> > > > pre-training on lots of simulation data
> > >
> > > We agree that it would be very nice to have a large amount of simulation pre-training data! However, we are not aware of any diverse simulated robotic pre-training datasets near the scale of Ego4D: most are constrained to a limited number of scenes, objects, and motions. Pre-training on Ego4D and evaluating in simulation is an established practice from prior works ( VIP (Ma et al 2022) and  R3M (Nair et al 2022)).

---

### Official Review · Reviewer_TboZ · 2024-10-27

**Soundness:** 3
**Presentation:** 2
**Contribution:** 2
**Rating:** 5
**Confidence:** 3

**Summary:**

This paper proposed a novel intrinsic reward method for sparse reward RL tasks. The intrinsic reward is derived from a goal-conditioned value function that is trained on Internet-based task-unrelated videos. The authors proved that the proposed ICVF could empirically help task training and performance. The method could promote RL's application in real-world tasks.

**Strengths:**

1. Although leveraging knowledge in internet large-scale videos and intrinsic reward are not fresh ideas, the authors design a framework that combines them and benefits sparse-reward RL.

**Weaknesses:**

1. This paper lacks theoretical analysis, especially the convergence of policy improvement and policy optimality after adding the intrinsic reward term.
2. What's the performance if removing all the sparse environment rewards?
3. Could you provide more comparisons with related baselines like FAC(Foundation Actor-Critic)?

**Questions:**

See above.

---

> ### Author Response · Authors · 2024-11-23
>
> Thank you for your thoughtful review. In response to your review, we provide ablation experiments in the Appendix, reference to theory in the main paper and Appendix A.2, and a baseline experiment against DreamerV2.
>
> **Theoretical analysis:** We provided additional analysis in reference to policy invariance to potential-based shaping (Ng. et al, 1999) in Appendix A.2 and Section 4.1 as well as reference to soft-policy improvement (Haarnoja et al, 2018) in Section 4.2 and implicit Q-Learning convergence (Kostrikov et al 2021) in Section 4.3. In relation to using a value vs. potential-based intrinsic reward, we conduct an experiment in AntMaze-Large-Diverse-v0 observing that the potential-based method exhibits higher variance returns shown in Figure 9 in Appendix A.2.
>
> **Sparse Reward Ablation:** We add an experiment in Appendix A.2 Figure 9 where we remove the extrinsic reward from the environment in Antmaze-Large-Diverse-v0.
>
> **Baseline:** We add experiments using Dreamerv2 (Hafner et al 2022),  a strong visual online RL algorithm. These results are added to Figure 3 and 5.

---

> > ### Comment · Reviewer_TboZ · 2024-11-25
> >
> > I think the authors' responses didn't resolve my concerns, so I will maintain my score.

---

> > > ### Author Response · Authors · 2024-11-28
> > >
> > > We are sorry to hear that our response did not resolve your concerns – could you please share what specific concerns remain? In our response we attempted to address the three areas of concern you noted in your original review (citing and discussing prior theoretical analysis, adding an ablation of extrinsic signals, and adding related baselines). We would love to work towards resolving other issues you found deficient.
> > >
> > > 1. For our algorithmic analysis, we wrote about design choices related to the inclusion of a shaping reward based on its theoretical and empirical outcomes as detailed in Appendix A.2.  We trade-off the usage of potential-based reward and value-based reward in Figure 11. We additionally include direct reference to convergence of IQL and convergence of SAC, the RL algorithms used for ICVF training and online RL, respectively.
> > > 2. Towards your question on removal of sparse environment reward, we conducted an experiment in Antmaze where we ablated all extrinsic reward signals. We found that removal of this reward did not significantly affect our algorithm’s performance as shown in Figure 10 (middle).
> > > 3. We have also added a baseline (Dreamerv2) that uses prior data in a different way from ours, through training a model and generating synthetic rollouts. Unfortunately, we were unable to evaluate the FAC method you suggested, since the code for this paper has not yet been released (https://github.com/YeWR/RLFP).

---

> > > > ### Comment · Reviewer_TboZ · 2024-12-02
> > > >
> > > > I appreciate the significant modifications authors have made to this paper. The writing has been improved.
> > > > 1. I apprecite your efforts on providing theoretical analysis, however, I think the authors should provide more details about Ng et. al (1999).
> > > > 2. I think one task is not enough to support the claim that removing environmental rewards will not affect training.
> > > > 3. I won't require the comparisons to FAC. But as reviewer U8sC mentioned, I cannot understand why you don't provide the Dreamer-v3 results.
> > > > I will maintain my score based on these considerations.

---

> > > > > ### Author Response · Authors · 2024-12-03
> > > > >
> > > > > Thank you for your response and understanding. We ran our experimental suite with DreamerV3 and observed that DreamerV2 performs better. We have now specified this in our paper and kept the DreamerV2 results for a better baseline comparison.
> > > > >
> > > > > > I think one task is not enough to support the claim that removing environmental rewards will not affect training.
> > > > >
> > > > > We'd be happy to add these to our paper. We currently remove any claims of having ViVa being unaffected by removal of extrinsic reward in the general case. We do expect the extrinsic reward to play the role of "locking the policy in place" during exploration as it is very sparse whereas ViVa guidance is smooth.
> > > > >
> > > > > > I apprecite your efforts on providing theoretical analysis, however, I think the authors should provide more details about Ng et. al (1999).
> > > > >
> > > > > Thank you. We add more reference to this paper with direct addition of the policy sufficiency proof for which we make direct usage of.

---

### Official Review · Reviewer_YvNk · 2024-11-04

**Soundness:** 3
**Presentation:** 3
**Contribution:** 3
**Rating:** 6
**Confidence:** 4

**Summary:**

This paper introduces a method called “ViVa” for pre-training intent-conditioned value functions (ICVF) on video datasets and then using the resulting value function to guide online RL in sparse reward settings. To do so, the authors define a goal-conditioned reward function that is -1 for all states except the goal state, where it is 0. Based on this, they then define an ICVF which models the unnormalized likelihood of reaching some outcome state from some starting state while following an optimal goal-reaching policy for a goal (intent) state. By setting the outcome state equal to the goal state they obtain a value function corresponding to the (negative) temporal distance to the goal state.

The authors pretrain this model on 3000 hours of internet video data, finetune it on environment-specific (but not task-specific) data, and then use the final value function to augment the environment’s reward function in online RL. They show that their approach performs favorably compared to several baselines (RLPD, SAC, JSRL).

**Strengths:**

- The paper makes a significant contribution to the approach of using internet-scale video data for pre-training RL models in order to improve the performance/data-efficiency of downstream RL. The comparison to baselines is fair and the experiments convincingly demonstrate the effectiveness of ViVa. I appreciate that the authors specifically included experiments to evaluate generalization and scaling properties of ViVa, in addition to evaluating performance.
- The related work section is generally comprehensive (with one caveat discussed below).
- The discussion section was very interesting, I particularly liked:
  - the comparison between ViVa and imitation policies w.r.t. generalization
  - and the proposed method for resolving value errors by collecting counterfactual examples with an exploratory policy
- The paper is well written and the presentation is good.

**Weaknesses:**

I would be happy to raise the score pending clarification/discussion of some of the concerns below.

**1. Discussion of imitation learning methods:**

The chosen baselines as well as the related work section focus on methods for learning value functions from videos. One closely related line of work that is omitted (both as a baseline as well as in the RW) is the approach of learning policies directly from video data, e.g. with latent policy imitation methods such as ILPO (Edwards et al. 2019), LAPO (Schmidt et al., 2023), Genie (Bruce et al., 2024), LAPA (Ye et al., 2024).

Relatedly, the authors claim
> Our analysis of ViVa illustrates the importance of using value function pre-training on video data as
opposed to other methods of utilizing prior data.

> Comparisons with JSRL further depict the superiority of value functions as a representation of prior
data as opposed to imitative policies.

I don't believe these claims are sufficiently supported by their results since (1.) they are based on a single method for imitative policies, JSRL, and (2.) crucially this method does not actually use the prior video data (Ego4D). The authors also state:
> Furthermore, imitative policies prevent support from action-free data sources such as Ego4D.

However, this limitation only applies to classical imitation learning methods (and JSRL), but not to methods for latent imitation from video.

Overall, based on the results shown, I believe the authors could argue for the importance of leveraging prior action-free video data, but would not be able to conclude that value function learning methods are generally superior to policy imitation methods.

**2. Other environments:**

The paper includes experiments on several robotics control environments (from RoboVerse, FrankaKitchen, AntMaze). I would be very interested in results from non-robotics environments, although I understand if that is beyond the scope of this paper.

**3. Minor issues:**

- Figure 1 is unclear: this is the first point where `s`, `s'`, `s+`, and `g` are used in the paper, but they haven't been defined yet.
- In the Figure 1 caption and also Section 4 it's a bit unclear what the difference between "future outcome", "intent", and "goal" is, plus it wasn't immediately obvious that all of those are states.
  - If intent and goal is the same thing (as is suggested in Sec 4.1), it might be better to stick to one of the two terms
  - To standardize the terms, I'd suggest using `current state` (`s`), `next state` (`s'`), `outcome state` (`s+`), `goal state` (`g+`)

**Questions:**

1. Could you clarify how exactly you performed environment finetuning? From Section 4.3, it is not entirely clear whether finetuning works exactly the same as ICVF pre-training (just on a different dataset), or whether this uses some ground-truth environment rewards to update the VF?

---

> ### Author Response · Authors · 2024-11-23
>
> Thank you for your detailed and thoughtful review. In response to your review, we have updated the paper with additional related work, ablation experiments, edited claims, more consistent notation and nomenclature, and a remake of Figure 1.
>
> **Latent Imitation Learning:** Great point! We have added relevant citations and discussion on latent imitation methods in Section 2 - Related Work as well as Section 6 - Discussion. As you mention, latent imitation methods are another promising way to use video datasets to accelerate online RL. We have updated the related work to include discussion about these methods in Section 2 (lines 117-120), and we have tempered the statements in Section 6 – Discussion (lines 458, 462, 464-6) to mention that reward shaping is not the only approach to learning from these offline datasets.
>
> **Other Benchmarks:** As you mention, we believe that studies outside robotics/control benchmarks would indeed depart from the scope of the paper.
>
> **Notation:** We edit the paper to mainly utilize “goal” rather than “intent” and reformat Figure 1 to more clearly depict our method pipeline.
>
> **Minor:**
> > Environment finetuning
>
> No ground truth reward is used for finetuning, we simply train the ICVF identically on the environment data. We have added this clarification in Section 4.3

---

> > ### Comment · Reviewer_YvNk · 2024-11-25
> >
> > Thank you, I greatly appreciate the clarification regarding my concerns as well as the updates to the paper. My concerns have been mostly addressed, but I would prefer to keep the same score. I believe ViVa is a valuable contribution and while the experiments generally appear rigorous & comprehensive, the results only slightly set ViVa apart from the best performing baseline in each case. As such I do lean towards acceptance but am not sufficiently confident to rate it as "8: accept, good paper".

---

### Official Review · Reviewer_U1Y2 · 2024-11-04

**Soundness:** 2
**Presentation:** 2
**Contribution:** 2
**Rating:** 3
**Confidence:** 4

**Summary:**

This submission proposes ViVa (Video-trained Value Functions), a method that learns goal-conditioned value functions from diverse video data (from Ego4D + environment-specific data) to develop an intent-conditioned value function (Ghosh et al., 2023) to guide the agent during online RL in sparse-reward settings.

**Strengths:**

- The paper is well-motivated and easy to understand.
- I appreciate that the authors used multiple seeds for their experiments and clearly stated the hyperparameters for all experiments to ensure reproducibility.

**Weaknesses:**

The paper, in its current form, lacks in certain areas outlined below:

1. While the authors report results over multiple seeds, the plots do not show standard deviation or error bars. This makes it difficult to assess the algorithm’s sensitivity to underlying hyperparameters and to compare the approach fairly with other algorithms like JSRL in Fig. 4 or RLPD in Fig. 5. I urge the authors to also show standard dev. / error for a fair comparison.
2. The presentation of all plots and figures could be improved by increasing the font size of the x- and y-axis ticks, as well as by enlarging the images in Figure 7. Currently, these are not readable in printed form or even when zoomed in on a laptop screen.
3. When reporting rewards in Figures 4 and 5, do the authors run the agent for multiple trials in the evaluation environment (e.g. 100 trials) and then report the mean or median? Or is the agent run only once? These details are necessary to support the SOTA performance claims.
4. Equation 4 combines the equations for loss to be minimized with the advantage function being estimated. Separating these two components would improve readability.
5. No training curves are provided for the Ego4D video pre-training or the online RL fine-tuning phases. Additionally, the exact pre-processing steps for Ego4D videos should be specified to enhance the reproducibility of the approach.
6. Minor: There is no discussion of computational costs or training times.


Overall, while the approach (ViVA with Ego4D) appears slightly more sample-efficient than other baselines, the results do not seem particularly promising. I would like to see details on the amount of effort (in terms of time and computational resources) required for pretraining on Ego4D, followed by fine-tuning, and finally training the agent for ViVA as well as the training time for RLPD and JSRL. If the effort is substantial, then the gains do not seem meaningful enough, as it appears that using Ego4D makes only a slight difference in the downstream task.

Therefore, I recommend a rejection but would be willing to reconsider my score based on the authors' rebuttal.

**Questions:**

1. On page 4, lines 201–205, the authors mention using a monolithic function instead of a multilinear formulation to learn the ICVF network. Are there any empirical results that the authors could share for this choice on at least one of the environments?
2. On page 5, in the *Guided Online RL* section, the authors state that they use 50% of online data from the replay buffer and 50% sampled from the prior dataset. Is there an ablation where the authors sampled only from the replay buffer (i.e., 100%) without any prior data?
3. Since the Ego4D dataset can include videos from multiple viewpoints, do the authors apply any pre-processing to make the ICVF agnostic to camera angle?
4. On page 7, line 348, it says "Appendix 2," but it should be corrected to "Appendix 8.2."
5. Typo:
    - Page 4, Line 206: “we working” → “we **are** working”

---

> ### Author Response · Authors · 2024-11-23
>
> Thank you for your detailed and thoughtful review. In response to your review, we have updated the paper with additional experiments, details about our experimental setup, in particular for the Ego4D pre-training. We politely disagree that pre-training on Ego4D is “not promising” – it yields considerable benefits in the low prior robot data setting (Fig 6, improves by 2x), is not especially expensive to acquire (1.5 days on a single TPU), and provides one way of incorporating broad external knowledge into downstream RL finetuning. Please see below for specific responses to your questions and concerns.
>
> **Ego4D Pre-Training Details, Metrics, Computational Cost:** We have updated Appendix A.1 with more details about the Ego4D pre-trainings setup, including Figure 8, which logs metrics through pre-training. This Ego4D pre-training stage is not very long (3M gradient steps @ batch size 64, takes ~36hrs on a v4-8 TPU) – in relative terms, finetuning for 1 seed of a single downstream task takes ~3hrs. Please see Section 5.2 in the paper for added details of the compute used in this paper. Our pre-training setup is equivalent in training split and pre-processing to other works pre-training on Ego4D, including V-PTR (Bhateja et al), VIP (Ma et al), R3M (Nair et al).
>
> **Multiple Seeds / Standard Deviation:** We have updated Figure 3, 4, 5 to include traces for every seed, to demonstrate the relative variabilities of all methods. Note that all the tasks are sparse reward tasks, so there is a nontrivial element of variability in when reward is first attained.
>
> **Ablation between Monolithic and Multilinear ICVF:** We have trained value functions on the full AntMaze-Large-Diverse dataset for both the multilinear and monolithic architectures, and visualized them in Figure 10 in Appendix A.3. Notice that while both versions coarsely capture the true value function in the environment, the multilinear one has strong artifacts / patterns in the predicted values, failure modes that the monolithic seems to not exhibit.
>
> **Ablation for Replay Buffer Only:** We conducted an ablation experiment with 0% (no prior data) sampling in the AntMaze-Large-Diverse-v0 domain. We observed that prior data sampling improves sample efficiency, although having no sampling still allows for solving the task. Please see Figure 9 in Appendix A.2 for a plot.
>
> **Minor:**
>
> > Improving Plot Fonts and Figure 7:
>
> We have updated to make Figure 1, 3, 4, 5, and 7 more readable.
>
> > Do the authors run the agent for multiple trials in the evaluation environment?
>
> When reporting, we take the mean across 10 evaluation episodes. We have added this detail to the caption in Figure 5 and 3.
>
> > Since the Ego4D dataset can include videos from multiple viewpoints, do the authors apply any pre-processing to make the ICVF agnostic to camera angle?
>
> No, we do not apply any pre-processing other than random-cropping. We have updated Appendix A.1. with exact information for Ego4D training.
>
> > Computational Cost
>
> The pre-train time was 1.5 days on a v4-8 TPU
> The fine-tune time for Hinge-task FrankaKitchen is 2.5 hours on a v4 TPU.
> The fine-tune time for Slide-task FrankaKitchen is 2.5 hours.
> The fine-tune time for COG on all available tasks is 9 hours.
> The fine-tune time for COG on the scaling law is 5 hours.
> The fine-tune time for AntMaze was 45 minutes on a Tesla V100.
>
> > Equation 4 combines the equations for loss to be minimized with the advantage function being estimated. Separating these two components would improve readability.
>
> We have edited the equation to separate out the lines.

---

> > ### Comment · Reviewer_U1Y2 · 2024-11-24
> > **Thanks for rebuttal**
> >
> > I thank the authors for answering my questions and updating the submission accordingly. I also appreciate that the authors added the plotlines for all seeds in Figures 3,4,5 as requested, however, the way the data is presented it is not at all understandable to me how much overlap there is between the baselines . I would like to see the authors either showing me error bars or something like this for better analysis between different baselines: https://stackoverflow.com/questions/12957582/plot-yerr-xerr-as-shaded-region-rather-than-error-bars or simply a table which states the mean and standard error / deviation. I am still not convinced regarding the novelty of this approach and also agree with Reviewer U8sC's remark *"If not using the Internet-scale datasets, I do not see how this work separates from other works on imitation learning."* Therefore, I am keeping my score of reject.

---

> > > ### Author Response · Authors · 2024-11-28
> > >
> > > Thank you for your response to our revision. To summarize our updates towards your concerns, we added pre-training information, computational costs across all experiments, additional visualizations of multilinear vs. monolithic VF’s, updated result figures, added an ablation experiment of prior data, improved mathematical readability, and responded to each of your questions. In our new revision, we have now also added a table of means and standard deviations for all our experiments in Appendix Table 2. We would like to know which concerns of yours are left unresolved in order to best improve our paper.
> > >
> > > Regarding your comments on method novelty, we would like to note that our work does differ from the imitation learning setup, even without usage of Internet-scale data. Our method makes no assumptions about optimality in the robot fine-tuning data, which is necessary for imitation learning methods. This is reflected in our experiments: all the offline robot data in COG consists of failures and partial trajectories, and similarly in FrankaKitchen, the trajectories have significant action noise and the presence of failures.

---

### Author Response · Authors · 2024-11-23

We would like to thank the reviewers for detailed commentary on our work! In summary, we have edited (shown in blue in the PDF revision) all figures for visibility, recreated Figure 1, and improved writing in the Methods Section. We have added a new DreamerV2 (Hafner et al 2022) baseline in Figures 3 and 5 along with additional comparisons to V-PTR (Bhateja et al 2023) in Appendix A.4. We have added reference to imitation methods in our Related Works / Discussion Section along with theoretical justifications of our design in Appendix A.2 and Methods Section. We also added more training configuration and compute details for pre-training (Figure 8) along with 3 of ablation experiments (Figure 9) and visualizations (Figure 10) in the Appendix. Lastly, we have addressed specific reviewer comments below.

---

### Meta-Review · Area_Chair_FrhL · 2024-12-06

**Metareview:**

This work proposes a method called ViVa that uses internet-scale video samples to learn a value function that encodes
goal-reaching priors. Then it is used in online RL and shows improvement. After rebuttal and discussion, this work receives the review scores of 3,3,5,6.

Reviewers raised several issues on (1) all simulated environments and no real-world experiments. (2) unclear math formulation and a lack of theoretical analysis. (3) lack of comparisons with more baseline. (4)  No analysis of the algorithm's sensitivity and many experimental issues.

The authors have prepared a rebuttal and addressed some of the concerns raised by the reviewers. But many remain. For example, there is still no real world experiment, so it is hard to tell if the manipulation task really works in real world. AC has checked the submission, the reviews, the rebuttal, and the discussions, and sided with the negative reviewers. Thus AC recommended rejection.

**Additional Comments On Reviewer Discussion:**

There is a rebuttal and discussion between reviewers and the authors. The reviewers have given detailed reasons why their scores mostly remain the same and stand on the rejection side. Even the reviewer with 6 acknowledged that "the results only slightly set ViVa apart from the best performing baseline in each case. As such I do lean towards acceptance but am not sufficiently confident to rate it as "8: accept, good paper"." Thus, a rejection is recommended by AC.

---

### Decision · Program_Chairs · 2025-01-22

Reject